# Functional Role of Arrestin-1 Residues Interacting with Unphosphorylated Rhodopsin Elements

**DOI:** 10.3390/ijms24108903

**Published:** 2023-05-17

**Authors:** Sergey A. Vishnivetskiy, Liana D. Weinstein, Chen Zheng, Eugenia V. Gurevich, Vsevolod V. Gurevich

**Affiliations:** Department of Pharmacology, Vanderbilt University, Nashville, TN 37232, USA; sergey.vishnivetskiy@vanderbilt.edu (S.A.V.); liana.weinstein@vanderbilt.edu (L.D.W.); chen.zheng@vanderbilt.edu (C.Z.); eugenia.gurevich@vanderbilt.edu (E.V.G.)

**Keywords:** arrestin, structure function, GPCR, receptor binding, protein–protein interactions

## Abstract

Arrestin-1, or visual arrestin, exhibits an exquisite selectivity for light-activated phosphorylated rhodopsin (P-Rh*) over its other functional forms. That selectivity is believed to be mediated by two well-established structural elements in the arrestin-1 molecule, the activation sensor detecting the active conformation of rhodopsin and the phosphorylation sensor responsive to the rhodopsin phosphorylation, which only active phosphorylated rhodopsin can engage simultaneously. However, in the crystal structure of the arrestin-1–rhodopsin complex there are arrestin-1 residues located close to rhodopsin, which do not belong to either sensor. Here we tested by site-directed mutagenesis the functional role of these residues in wild type arrestin-1 using a direct binding assay to P-Rh* and light-activated unphosphorylated rhodopsin (Rh*). We found that many mutations either enhanced the binding only to Rh* or increased the binding to Rh* much more than to P-Rh*. The data suggest that the native residues in these positions act as binding suppressors, specifically inhibiting the arrestin-1 binding to Rh* and thereby increasing arrestin-1 selectivity for P-Rh*. This calls for the modification of a widely accepted model of the arrestin–receptor interactions.

## 1. Introduction

Most cellular functions are regulated by protein–protein interactions. As a rule, a binding partner prefers a particular conformational form of its target. In addition, the interaction is often triggered or enhanced by certain post-translational modifications in one or both binding partners. The engagement of G protein-coupled receptors (GPCRs) by their binding partners demonstrates both modes of regulation. G proteins and GPCR kinases (GRKs) preferentially bind their cognate receptors in the active conformation. Arrestins bind active and phosphorylated GPCRs with significantly greater affinity than all other functional forms of the same receptor (reviewed in [1]). Visual arrestin-1 (note that we use systematic names of arrestin proteins, where the number after the dash indicates the order of cloning: arrestin-1 (historic names S-antigen, 48 kDa protein, visual or rod arrestin), arrestin-2 (β-arrestin or β-arrestin1), arrestin-3 (β-arrestin2 or hTHY-ARRX), and arrestin-4 (cone or X-arrestin)). selectively binds light-activated phosphorylated rhodopsin (P-Rh*), while demonstrating much lower binding to the inactive phosphorylated and active unphosphorylated (Rh*) rhodopsin, and virtually no binding to the inactive unphosphorylated form [2]. The original model explaining this selectivity posited that arrestin has two structural elements that act as independent sensors, the phosphate and active receptor sensors, which only P-Rh* can engage at the same time [3]. The model postulates that simultaneous engagement of these two sensors promotes arrestin transition into a high-affinity receptor-binding conformation, which brings additional elements into contact with the receptor, thereby increasing the energy of the interaction, and arrestin affinity [3].

Arrestin residues involved in phosphate binding have been extensively studied by mutagenesis [4,5,6,7,8,9]. The key residues in the phosphate sensor were identified in visual arrestin-1 [9,10,11] and both non-visual subtypes [7,9,12,13,14]. The role of the “finger loop” (term from [15]) (Figure 1) as the activation sensor has recently been established in arrestin-1 [16] and non-visual arrestins [17,18]. The role of arrestin-1 residues that are not part of either sensor was only tested in a screen of their alanine substitutions [19,20]. In these experiments, ~44 kDa arrestin-1 was tagged with very large 28 kDa mCherry. Moreover, different concentrations of mutants (which depended on their expression levels in *E. coli*) were used, so the absolute binding levels could not be numerically compared. Instead, the authors compared NaCl sensitivity of the binding, not the binding values [19,20]. The role of these residues in a direct rhodopsin binding assay involving two wild type (WT) proteins has not been tested (with the exception of the middle loop [21]).

In all arrestin–receptor complex structures solved so far, various “enhanced” mutants, not WT arrestins, were used to increase the complex stability [11,22,23,24,25,26,27,28]. A recent mutagenesis study suggested that the sets of residues involved in the receptor binding by WT arrestin-1 and its enhanced mutants overlap only partially [21]. Here we examined the role that the elements of WT arrestin-1 contacting rhodopsin in the crystal structure of the complex (Figure 1) [11,22] play in the rhodopsin binding. We probed by mutagenesis the C-loop (residues 249–254 in bovine arrestin-1), which interacts with the intracellular loop 2 of rhodopsin; the back loop (residues 281–322), in which its C-terminal part (residues 316–322) binds the C-terminus of rhodopsin and transmembrane helix (TM) V [22]; and β-strand VI (residues 82–89) that precedes the finger loop. To compare the role of these residues in the receptor interaction of WT and enhanced form of arrestin-1, we introduced the same mutations in the context of both WT protein and its truncated (1–378) mutant (Tr).

## 2. Results

The common assumption in the systematic mutagenesis studies of protein–protein interaction interfaces is that if a particular residue participates in the binding of one protein to another, the mutations in that position, particularly non-conservative ones (such as charge reversals), weaken the interaction. If the interface is extensive (in the crystal structure of the complex the arrestin-1–rhodopsin interface is ~1350 Å^2^ [22], which is quite extensive) and involves multiple side chains, a mutation affecting only one residue might not produce a dramatic effect, but it is still expected to be detrimental for the interaction to some extent. This “linear” thinking is not supported by our data on arrestin-1 binding to rhodopsin, suggesting that many residues in arrestin-1 play a regulatory role.

### 2.1. β-Strand VI

In the structure of its complex with rhodopsin, *β*-strand VI of arrestin-1 (nomenclature from [15]) makes extensive contacts with the cytoplasmic side of TM5 and TM6, as well as with the intracellular loop (ICL) 3 connecting TM5 and TM6, accounting for one of the four patches of the interface [22] (Figure 1B and Figure 2B). We introduced ten mutations into arrestin-1 β-strand VI, including one charge reversal (Asp82Arg) (Figure 2A). The most remarkable and rather unexpected result is that none of these mutations reduced arrestin-1 binding to its preferred form of rhodopsin, P-Rh* (Figure 2C). In fact, the most drastic substitution Asp82Arg, which changed negatively charged aspartic acid into positively charged arginine, significantly increased P-Rh* binding (Figure 2C). Alanine substitutions of Leu83, Ser86, and Gln89 also resulted in increases in P-Rh* binding, although much smaller than Asp82Arg (Figure 2C). The binding of four mutants on the WT background to unphosphorylated Rh* was also increased, most significantly upon substitutions of Asp82 (Figure 2C). However, the absolute level of WT arrestin-1 binding to Rh* is much lower than to P-Rh* (Figure 2B,C), which makes numerical comparisons less reliable than in the case of P-Rh*.

The C-terminus of arrestin-1 is detached from the body of the molecule upon rhodopsin binding [29,30,31,32]. The deletion of the arrestin-1 C-terminus [8,9,16,21,33], as well as its forcible detachment by alanine substitutions of the bulky hydrophobic residues that anchor it to the N-domain [8,21,33], yields mutants with greatly increased binding to Rh*. Therefore, to gain additional insight into the effect of mutations on Rh* binding, we used C-terminally truncated arrestin-1-(1–378) (Tr). WT arrestin-1 and its enhanced mutants appear to engage P-Rh* differently; the binding of enhanced mutants involves residues that do not significantly contribute to WT arrestin-1 binding [21]. Therefore, we tested the binding of β-strand VI mutants on the Tr background to both P-Rh* and Rh*. The binding of Tr to P-Rh* was more sensitive to β-strand VI mutations than that of WT: the Phe85Ala mutation increased it, while three others were detrimental (Asp82Arg, Val88Ala, and Gln89Ala) (Figure 2E). High binding of Tr to Rh* was further enhanced only by Gln87Ala substitution (Figure 2F). It was significantly decreased by five mutations: Asp82Ala, Asp82Arg, Val88Ala, Gln89Ala, and most dramatically by Leu83Ala (Figure 2F). Asp82Arg, Val88Ala, and Gln89Ala decreased Tr binding to both P-Rh* and Rh*, suggesting that these three residues likely participate in the interaction of this form of arrestin-1 with both forms of rhodopsin. In contrast, Leu83Ala and Gln87Ala affected Rh* binding of Tr, but not P-Rh* binding, suggesting that these residues are involved in the interaction of truncated arrestin-1 only with unphosphorylated rhodopsin.

The opposite effects of D82R mutation on P-Rh* binding, an increase in the case of WT and a decrease in the case of Tr, suggest that aspartic acid in this position in the WT protein likely participates in maintaining the basal conformation of arrestin-1. In that case, its replacement facilitates rhodopsin binding in the conformationally constrained WT arrestin-1. This function becomes irrelevant in the Tr mutant, which is already conformationally “loosened up” by the deletion of the C-terminus, so that the substitution reveals a direct role of this residue in the rhodopsin interaction (at least in the case of the Tr; the data do not mean that this residue also mediates the binding of WT arrestin-1 to P-Rh*). This residue apparently participates in the binding of the Tr mutant to the unphosphorylated rhodopsin elements, as Asp82Arg significantly reduces its binding to Rh*, which does not carry attached phosphates. Indeed, in the crystal structure of the complex (where the enhanced arrestin-1 mutant with a triple alanine substitution that detaches its C-terminus (3A mutant) was used), Asp83 (in mouse arrestin-1, homologous to Asp82 in bovine protein used here) contacts Gln237 in the ICL3 of rhodopsin [22].

### 2.2. The C-Loop

In the structure of the arrestin-1–rhodopsin complex [22], the C-loop, along with a recently explored middle loop [21], makes one of the “patches” of the arrestin–rhodopsin interface. In particular, Tyr251 (in mouse arrestin-1, homologous to Tyr250 in bovine protein) contacts Cys140 in TM3 and Thr229 in TM5 [22] (Figure 3B). To determine the role of this arrestin-1 element in rhodopsin binding, we introduced a series of mutations in it (Figure 3A) and tested the binding of the mutants on the WT background to P-Rh* (Figure 3E) and Rh* (Figure 3F). Leu249Ala and Y250Ala slightly reduced P-Rh* binding, in which arrestin-1 interaction with rhodopsin-attached phosphates plays an important role (reviewed in [1]). The same two mutations and Ser251Ala reduced arrestin-1 binding to Rh* (Figure 3F). However, Leu249Ala and Tyr250Ala increased the binding of truncated arrestin-1 to P-Rh*, whereas these two mutations and Ser251Ala suppressed Tr binding to Rh* (Table 1) (Figure 3G,H). Thus, native residues in these positions appear to participate in the binding of WT arrestin-1 to both forms of rhodopsin and in Tr binding to Rh* but are detrimental for Tr binding to P-Rh*. Both substitutions of Asp253 severely reduced Tr binding but did not appreciably affect the binding of WT arrestin-1. The substitution Tyr254Ala affected Rh* binding of Tr but did not have any other effects. The opposite effects of Leu249Ala, Tyr250Ala, and Thr319Glu on P-Rh* and Rh* binding of Tr (Figure 3; Table 1) suggest that different residues of this form of arrestin-1 engage these two forms of rhodopsin.

### 2.3. Back Loop Residues That Contact Rhodopsin in the Structure

According to the structure of the complex [22], part of the back loop of arrestin-1 contacts Gln236 in the C-terminus of rhodopsin TM5 (Figure 3D), while Arg292 (Arg291 in bovine protein) contacts Met143 and Arg147 in rhodopsin ICL2 [22] (Figure 3C). Therefore, we tested how the substitutions of Thr319 in the back loop and Arg291 affect WT arrestin-1 binding to P-Rh* (Figure 3E) and Rh* (Figure 3F). While Thr319Ala mutation did not affect the binding to either form of rhodopsin, Thr319Glu reduced the binding of WT arrestin-1 to both forms and the binding of Tr to Rh*, while enhancing Tr-P-Rh* interaction (Figure 3; Table 1). Elimination of the side chain of Arg291 (Arg291Ala) significantly increased the binding of WT arrestin-1 to both P-Rh* and Rh* (Figure 3E,F). A positive effect of Arg291Ala on binding to both forms of rhodopsin was retained on the Tr background (Figure 3G,H). Thus, Arg291Ala mutation is the only one among 21 tested that produced the same effect on the binding of both forms of arrestin-1 to both functional states of rhodopsin, suggesting that WT arginine in this position suppresses the interaction in all cases. The increase in the WT arrestin-1 binding to P-Rh* is moderate, whereas this mutation increases the binding to Rh* severalfold. Thus, it appears that the functional role of Arg291 is to enhance arrestin-1 selectivity for P-Rh*.

## 3. Discussion

To date, site-directed mutagenesis is the only method of probing the interactions of WT arrestins with WT GPCRs [4,5,8,9,10,16,21,33,34,35]. In all available structures of the arrestin–receptor complexes mutationally enhanced arrestins, not the WT forms, were used to increase the complex stability. In particular, 3A arrestin-1 was crystallized in complex with rhodopsin [11,22]; solved structures contain truncated arrestin-2 bound to M2 muscarinic receptor [24], truncated cysteine-free arrestin-2 [26] or its 3A mutant [23] bound to the neurotensin receptor, enhanced R169E polar core mutant of arrestin-2 [13,14] bound to the β1-adrenergic receptor [25], truncated arrestin-2 (1–382) bound to the vasopressin V2 receptor [27], and doubly enhanced arrestin-2 (R169E polar core mutation plus the deletion of the C-terminus) bound to the 5HTB receptor [28]. Our recent study strongly suggested that WT arrestin-1 and its mutationally enhanced variants bind rhodopsin in distinct ways, apparently using only partially overlapping sets of residues to engage the receptor [21]. Thus, the insights from the solved structures may not be directly applicable to WT arrestin proteins.

Arrestin-1 binds light-activated phosphorylated rhodopsin, P-Rh*, with high selectivity, demonstrating many times lower binding to all other functional forms of rhodopsin, including Rh* that emerges upon rhodopsin activation by light [2,3]. Two sets of residues have been identified in the arrestin-1 molecule: the first referred to as the activation sensor recognizing the activated conformation of the receptor [16] and the other, the phosphorylation sensor, recognizing its phosphorylation state [8,9]. However, the crystal structure of the rhodopsin–arrestin complex revealed numerous arrestin-1 residues in contact with rhodopsin that do not belong to either sensor [11,22]. Here we explored the role of some of these residues in the arrestin-1 binding to rhodopsin by targeted mutagenesis. The residues targeted here were selected based on the crystal structure of enhanced arrestin-1-3A mutant bound to constitutively active rhodopsin mutant [22], as the structure of the biologically relevant complex of the two WT proteins is not available. Our data suggest that the enhanced Tr mutant binds rhodopsin similarly to the 3A mutant: many mutations on the Tr background suppressed binding, as could be expected if the residue in question directly participates in the process. The mutations of residues whose only role is direct interaction with the receptor are likely to decrease the observed binding. In contrast, mutations of the residues that affect the binding indirectly (e.g., reduce or increase the probability of arrestin-1 transition into the conformation favorable for the rhodopsin interaction) can change the binding in either direction. The overview of the statistically significant changes (Table 1) suggests that the structure obtained with the enhanced 3A mutant of arrestin-1 [22] predicts the mode of interaction of the enhanced Tr mutant used here much better (seventeen mutation-induced decreases and eight increases in binding) than the mode of interaction of the WT arrestin-1 (seven decreases and eleven increases in binding) (Table 1). If we only count mutations of the four residues conserved in all mammalian arrestin subtypes (Figure 4), which were substituted in five mutants, the decrease-to-increase ratio is also much greater for Tr (6:1) than for WT arrestin-1 (1:3). Thus, the data indicate that the mode of interaction of enhanced mutants with rhodopsin (3A in [11,22], Tr in this study) is different from that of the WT arrestin-1. Only one mutation (Arg291Ala) out of 21 tested produced the same effects on WT and Tr backgrounds (Figure 3, Table 1). It is hardly surprising that the reported structure of the rhodopsin complex with the 3A mutant [22] predicts the binding mode of the Tr mutant fairly well. For example, substitutions of Asp82 reduce the binding on the Tr background, suggesting that this residue participates in the interaction of the Tr form with rhodopsin, as it does in the case of the similarly enhanced 3A mutant in the crystal structure of the complex [22]. Notably, on the Tr background, several mutations changed binding to P-Rh* and Rh* in opposite directions (Figure 3, Table 1), suggesting that this enhanced arrestin-1 mutant employs different residues for the binding to these two forms of rhodopsin. Overall, our data suggest that the ability of the available structure to predict the residues of WT arrestin-1 engaged by rhodopsin is limited.

The mutagenesis approach employed here yielded important insights into the mode of the WT arrestin-1 interaction with rhodopsin. Mutations on the WT background invariably produced changes in terms of the binding to P-Rh* and Rh* in the same direction (Table 1). This suggests that in the case of the WT arrestin-1, the residues that are involved in or indirectly regulate the binding to Rh* play the same roles in P-Rh* interaction. Some mutations increase the binding of the WT arrestin-1 to both the P-Rh* and Rh*, others only to Rh*. A mutation-induced increase in the WT arrestin-1 binding indicates that the mutated residue is unlikely to interact with rhodopsin directly. All mutations that increase the binding of the WT arrestin-1 to P-Rh* increase its binding to Rh* to a much greater extent (Figure 2 and Figure 3). These findings suggest that the native residue in that position suppresses the binding to Rh*, thereby enhancing the arrestin-1 selectivity, in some cases even at the expense of somewhat reducing its binding to the preferred target, P-Rh*. Thus, the functional role of WT residues Asp82, Tyr84, Gln87, and Arg291 is to increase the arrestin-1 selectivity, i.e., its preference for P-Rh* over Rh*. The most parsimonious explanation of the effect of these mutations is that the native residues act indirectly, likely reducing the probability of the arrestin-1 transition into the binding-competent conformation. Conceivably, this can be achieved by increasing the energy barrier of this transition, so that a weak “push” of Rh* is not sufficient for the WT arrestin-1 to “jump” over the barrier. Naturally, a higher energy barrier would reduce the probability of this transition even upon a harder “push” provided by P-Rh*. It would still serve as an effective filter precluding the high-affinity binding to Rh*, i.e., its main biological function must be to increase the arrestin-1 selectivity for P-Rh*.

The identification of several residues that suppress arrestin-1 binding to a non-preferred form of rhodopsin, Rh*, along with earlier identification of the binding-suppressing WT residues in the middle loop [21], supports the hypothesis (first proposed in [21]) that arrestin-1 has residues that function as suppressors of Rh* binding, thereby further enhancing its selectivity for P-Rh*. This selectivity-enhancing mechanism functions in addition to the “coincidence detector” mechanism involving the two sensors independently responding to the receptor activation and phosphorylation envisioned by the original sequential multi-site binding model (proposed in [3], explained in detail in [1]).

Mammals have four arrestin subtypes [36]. Two of these, arrestin-1 and -4 (also known as rod and cone arrestins), are expressed in photoreceptors in the retina and bind photopigments. The other two, arrestin-2 and -3 (also known as β-arrestin1 and 2) are ubiquitously expressed and interact with the majority of non-visual GPCRs, i.e., with hundreds of different receptors. Visual arrestin-1 demonstrates remarkable selectivity for the preferred functional form of rhodopsin, P-Rh*, with its binding to Rh* and inactive phosphorylated rhodopsin being only 5–10% of the binding to P-Rh* [2,3,33]. In contrast, both non-visual subtypes are much less selective: the binding to the active phosphorylated forms of their cognate GPCRs is only 2–3-fold greater than to the active unphosphorylated or inactive phosphorylated forms [13,14,37]. However, all arrestin subtypes demonstrate certain selectivity, apparently employing similar mechanisms to achieve it [1]. Residues that play a role in the receptor binding mechanisms shared by all arrestin subtypes are expected to be conserved. In contrast, residues directly involved in the receptor interaction are less likely to be the same in the visual and non-visual subtypes because the receptor specificity of arrestin-1 and -2 is dramatically different [34,35]. Thus, it is instructive to compare the sequence of the elements tested in this study in all four mammalian arrestins (Figure 4). In β-strand VI, the first two and the last residues are conserved, the third residue is aromatic (Tyr in both visual and Phe in non-visual), and the fourth is hydrophobic (Phe, Val, or Ile), whereas the next three are not conserved (Figure 4). In the C-loop, the first residue is conserved, the second is aromatic (again, Tyr in visual and Phe in non-visual), and the third has a relatively small side chain with H-bonding capability (Ser or Asn), while the following three are not conserved (Figure 4). The residues homologous to Arg291 always have a positively charged side chain: it is Arg in arrestin-1 and Lys in the other three subtypes (Figure 4). Thr319 is not conserved: the other three arrestins have glutamic acid in homologous positions (this motivated us to change it to Glu in arrestin-1) (Figure 4). Interestingly, in the β-strand VI, the majority of functional changes (five out of eight on WT and six out of ten in Tr) occur due to mutations of conserved residues (Table 1). All substitutions of conserved residues (Asp82Ala, Asp82Arg, Leu83Ala, and Gln89Ala) invariably enhance the binding of the WT arrestin-1 but reduce the binding of Tr (Figure 2; Table 1). These data suggest that these mutations affect mechanisms ensuring selectivity that might operate in all WT arrestins but are “turned off” in enhanced mutants. Alanine substitutions of conserved Leu249 and semi-conserved Tyr250, as well as Thr319Glu mutation, reduce the binding of the WT arrestin-1 to both forms of rhodopsin, suggesting that these residues likely directly participate in the interaction. Arg291Ala mutation increased the binding of both full-length and Tr arrestin-1 to the two forms of rhodopsin (Figure 3). This suggests that the native Arg in this position suppresses the binding. Its replacement with alanine increased P-Rh* binding moderately, while more than doubling Rh* binding, suggesting that the most likely role of Arg291 is to increase arrestin-1 selectivity for P-Rh*. It is tempting to speculate that positively charged Lys in homologous positions of the other three arrestin subtypes has the same function. This should be tested experimentally.

The value of our data is two-fold. First, the results improve our understanding of the molecular mechanism of the WT arrestin-1 binding to P-Rh* and call for a refinement of a widely accepted model of the arrestin–GPCR interaction. Several arrestin residues that are not a part of either sensor apparently serve as suppressors of binding to the non-preferred forms of the receptor. This novel element must be added to the model. Second, the data are necessary to guide the construction of efficient Rh*-binding enhanced mutants to compensate for the defects in rhodopsin phosphorylation in human patients expressing mutant rhodopsin [38,39] or defective rhodopsin kinase [40,41,42]. Arg291Ala mutation that increased Tr binding to both P-Rh* and Rh* is a good candidate for inclusion in compensating enhanced mutants.

## 4. Materials and Methods

### 4.1. Materials

[γ-^32^P]ATP and [^14^C]leucine were purchased from PerkinElmer (Waltham, MA, USA). Restriction endonucleases and T4 DNA ligase were from New England Biolabs (Ipswich, MA, USA). Rabbit reticulocyte lysate was custom-made in bulk by Ambion (Austin, TX, USA). SP6 RNA polymerase was expressed in *E. coli* and purified, as described [43]. DNA purification kits for mini (3 mL) and maxi (100 mL) preparations were from Zymo Research (Irvine, CA, USA). All other reagents were from Sigma-Aldrich (St. Louis, MO, USA).

### 4.2. Mutagenesis and Plasmid Construction

For in vitro synthesis of corresponding mRNAs, bovine arrestin-1 was subcloned into pGEM2 vector (Promega; Madison, WI, USA) with “idealized” 5-UTR that does not require a cap for efficient translation [43] between Nco I and Hind III sites, as described [44]. Mutations were introduced by PCR. Appropriate unique restriction sites in the reengineered coding sequence of bovine arrestins-1 [35] were used to subclone generated fragments into this construct (Bam HI—Pst I for β-strand VI, Apa I—Bal I for the C-loop, and Bal I—Bst XI for the back loop). All mutations were confirmed by dideoxy sequencing (GenHunter Corporation, Nashville, TN, USA). Appropriate restriction fragments containing mutant sequence were excised from wild type (WT) constructs and subcloned into pGEM2-based construct encoding Tr mutant.

In vitro transcription, translation, calculation of specific activity of produced arrestin proteins, and preparation of different functional forms of phosphorylated and unphosphorylated rhodopsin were performed as described [9,16,44,45,46]. Briefly, all coding sequences with idealized 5′-UTR [43] were subcloned into pGEM2 vector (Promega, Madison, WI, USA) under control of SP6 promoter. Before in vitro transcriptions using SP6 RNA polymerase plasmids were linearized with Hind III (Hind III site is downstream of the stop codon). Uncapped mRNAs (idealized 5′-UTR obviates the need for cap) were produced, as described [43]. Cell-free translation of uncapped mRNAs in the presence of the mix of 19 unlabeled amino acids and [^14^C]leucine to generate radiolabeled arrestin-1 mutants in rabbit reticulocyte lysate was performed, as described [2,47]. Upon completion of translation, the mix was cooled on ice and centrifuged for 1 h at 100,000 rpm (>100,000× *g*) in TLA 100.1 rotor in Optima TLX tabletop ultracentrifuge (Beckman) to pellet ribosomes along with misfolded/aggregated proteins. Protein-incorporated radioactivity was determined, as described [47], before and after high-speed centrifugation. The fraction of the radiolabeled protein that remains in the supernatant after centrifugation was calculated and used as a criterion of proper protein folding and stability (it was 80–85% for WT arrestin-1 and all mutants used in this study).

### 4.3. Direct Binding Assay 

Direct binding assay was performed, as described [44,46]. Briefly, 1 nM arrestin-1 (50 fmol, specific activity 10.9–12.9 dpm/fmol) was incubated with 0.3 μg of indicated functional forms of rhodopsin (P-Rh* or Rh*) in 50 μL of 50 mM Tris-HCL, pH 7.4, 100 mM potassium acetate, 1 mM EDTA, 1 mM DTT for 5 min at 37 °C under room light. Samples were cooled on ice, then bound, and free arrestin-1 was separated at 4 °C by gel filtration on 2 mL column of Sepharose 2B-CL. Arrestin-1 eluting with rhodopsin-containing membranes was quantified by liquid scintillation counting on Tri-Carb (PerkinElmer, Waltham, MA, USA). Non-specific “binding” (likely reflecting arrestin-1 aggregation) was determined in samples without rhodopsin and subtracted.

### 4.4. Data Analysis and Statistics

Statistical significance was determined using one-way ANOVA (analysis of variance) with Dunnett’s multiple comparison test using GraphPad Prism software. *p*-values < 0.05 were considered statistically significant and indicated as follows: * *p* < 0.05; ** *p* < 0.01; *** *p* < 0.001.

## 5. Conclusions

Site-directed mutagenesis ([9,16,21] and this study) suggests that available structures of the receptor–arrestin complexes [11,22,23,24,25,26,27,28] reveal the mode of interaction of the enhanced arrestin mutants with engineered receptors used for structure determination, which is not necessarily the mode employed by WT proteins. The interaction of two WT or near-WT proteins can be explored by mutagenesis combined with a direct binding assay [2,3,4,5,8,9,10,14,16,19,20,21,33,34,35,37,44,45,48,49], by biophysical methods [30,31,32,50,51], or by in-cell cross-linking of the two proteins [52,53]. However, these methods have their own caveats. In the first two cases, proteins interact outside of the natural intracellular environment. In the latter case, the interaction occurs in the physiologically relevant environment of the living cell, but in this method (as well as when spin-labeled proteins are used for EPR), the proteins are close to WT but not exactly WT, as point mutations have to be introduced. Thus, there are no perfect methods to study the interaction of WT arrestins with native GPCRs, so the information gleaned by different methods must be integrated.

## Figures and Tables

**Figure 1 ijms-24-08903-f001:**
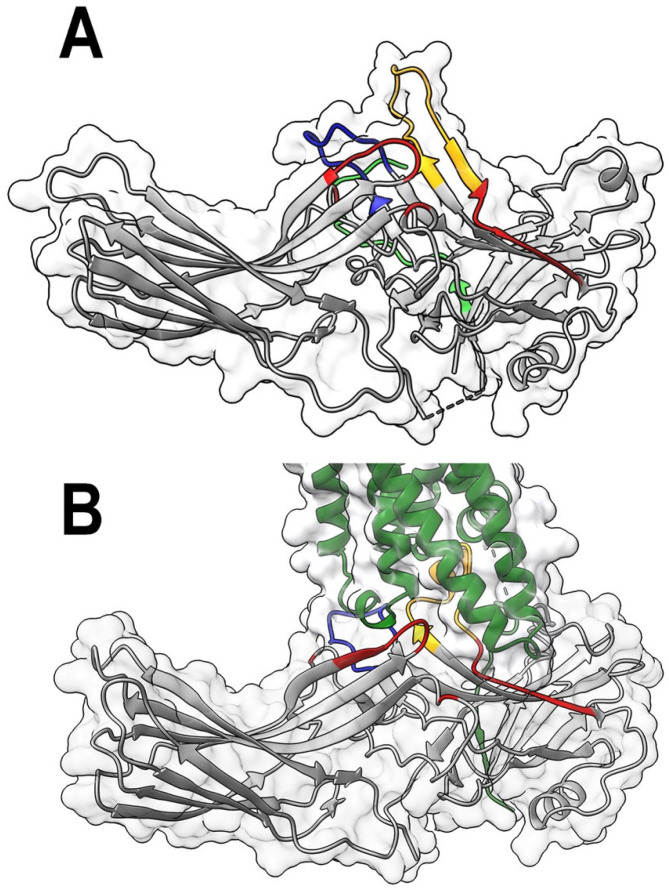
**The positions of targeted elements in free and rhodopsin-bound arrestin-1.** (**A**). Arrestin-1 (molecule A in the crystal tetramer of bovine arrestin-1, PDB ID: 1cf1 [15]) with residues mutated in this study shown in red, the finger loop (residues 68–81) in yellow, and the middle loop (residues 132–142) in blue. The attached part of the arrestin-1 C-terminus resolved in structure is shown in bright green. (**B**). The structure of the mouse arrestin-1 complex with rhodopsin (complex A, PDB ID: 5w0p [11]). Arrestin-1 (gray) in all panels and rhodopsin (dark green) in panel (**B**) are shown as flat ribbon with molecular surface of arrestin-1 indicated. The direction (N-to-C) of β-strands is shown by arrows. Note that residue numbers in mouse arrestin-1 compared to bovine arrestin-1 are N + 1. Images were created in DS ViewerPro 6.0 (Dassault Systèmes, San Diego, CA, USA).

**Figure 2 ijms-24-08903-f002:**
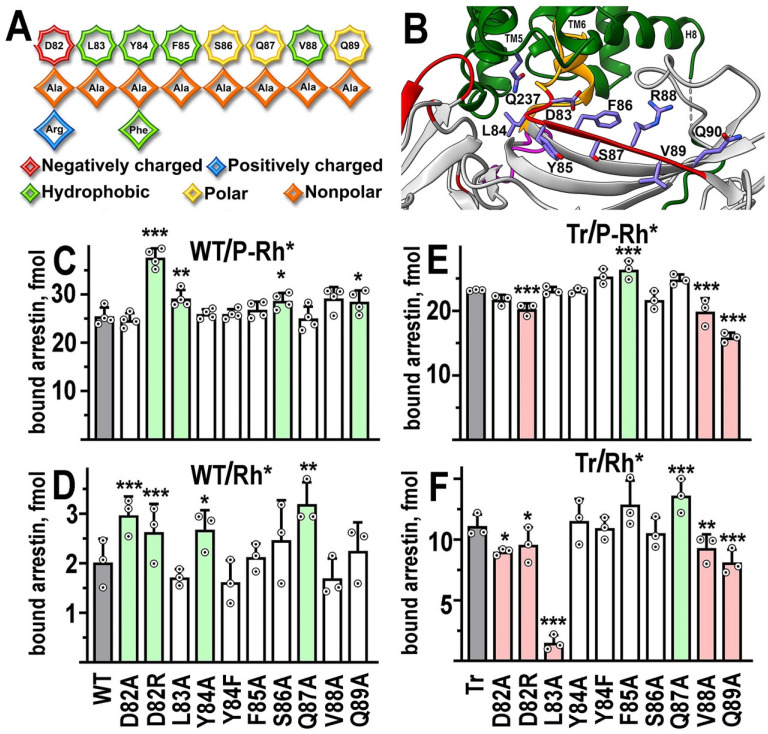
**The functional role of β-strand VI residues.** (**A**) The linear WT sequence of β-strand VI of bovine arrestin-1 is shown on top, with residues represented as octagons. Substituting residues used in this study are shown as rhombi. The chemical nature of the original side chain and its replacements is shown by color, as indicated. (**B**) The position of β-strand VI in the crystal structure of the arrestin-1 complex with rhodopsin [11,22]. The depiction of rhodopsin and arrestin-1 is the same as in Figure 1; the middle loop is shown in magenta, the rest of the color scheme is the same as in Figure 1. In addition, side chains of rhodopsin residue Gln237 and mouse arrestin residues 83–90 (homologous to bovine arrestin-1 residues 82–89) Asp-Leu-Tyr-Phe-Ser-Arg-Val-Gln are shown (note that in the bovine protein, Gln87 occupies the position of mouse Arg88). Residues are labeled using single letter code. Rhodopsin TM5, TM6, and helix-8 (H8) are indicated. The image was created in DS ViewerPro 6.0 (Dassault Systèmes, San Diego, CA, USA). (**C**–**F**) The binding of indicated mutants of WT arrestin-1 to P-Rh* (**C**) and Rh* (**D**), as well as the binding of truncated (Tr) arrestin-1-(1–378) with the same mutations to P-Rh* (**E**) and Rh* (**F**) was determined using radiolabeled arrestins produced in cell-free translation in the direct binding assay with purified phosphorylated or unphosphorylated light-activated bovine rhodopsin, as described in Methods. Full-length native bovine rhodopsin was used in all assays; WT and Tr refer to full-length arrestin-1 (1–403) and its truncated (1–378) form, respectively. Circles represent measurements in independent experiments, each performed in duplicate (n = 3–4). The binding was analyzed separately in each of the four groups. Statistical significance of the differences between parental WT (**C**,**D**) or Tr (**E**,**F**) (darker shaded bars in panels (**C**–**F**)) and mutants was determined by ANOVA followed by Dunnet post hoc test with correction for multiple comparisons and is indicated as follows: * *p* < 0.05; ** *p* < 0.01; *** *p* < 0.001. Bars showing statistically significant increases and decreases in binding are shaded light green and pink, respectively.

**Figure 3 ijms-24-08903-f003:**
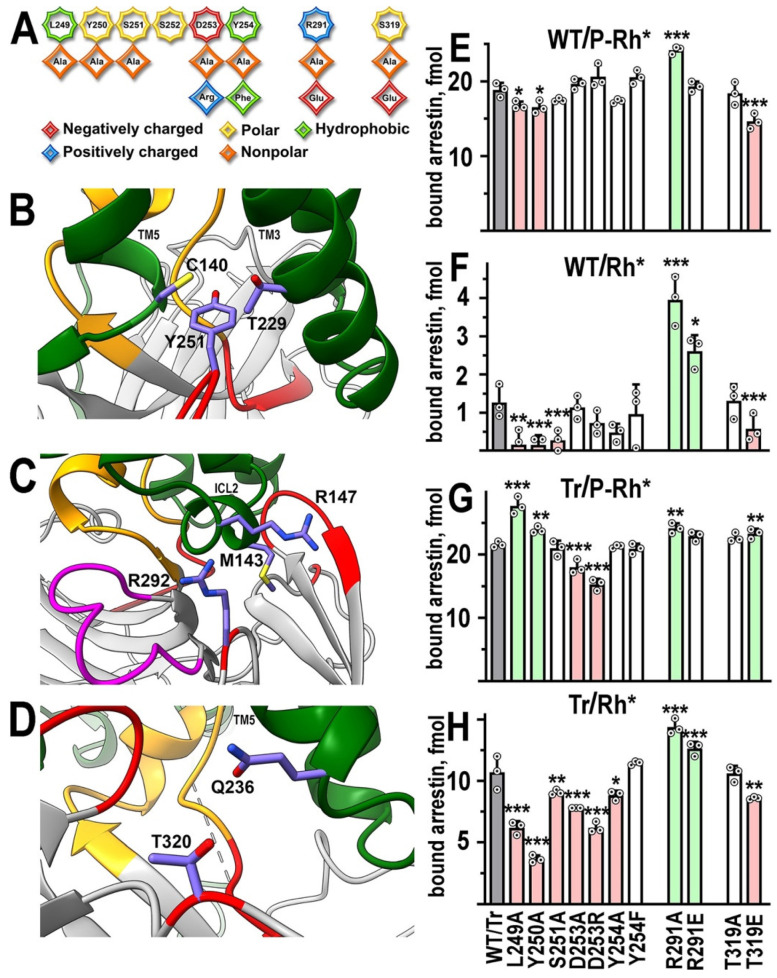
**Functional role of rhodopsin-contacting residues in arrestin-1.** (**A**) Linear WT sequence of the C-loop and other residues mutated in this study of bovine arrestin-1 are shown on top, with residues represented as octagons. Substituting residues used in this study are shown as rhombi. The chemical nature of the original side chain and its replacements is shown by color as indicated. (**B**–**D**) The positions of targeted arrestin-1 elements in the crystal structure of its complex with rhodopsin [11,22]. The depiction of rhodopsin and arrestin-1 is the same as in Figure 1; the middle loop is shown in magenta, the rest of the color scheme is the same as in Figure 1. In addition, the side chains of rhodopsin residues Cys140 and Thr229 are shown in (**B**), Met 143 and Arg147 in (**C**), and Gln236 in (**D**). Mouse arrestin-1 residues (corresponding numbers of bovine arrestin-1 residues are N-1) Tyr251 (**B**), Arg 292 (**C**), and Thr320 (**D**) are also shown. Residues are labeled using single letter code. Rhodopsin elements TM3 and TM5 (**B**), ICL2 (**C**), and TM5 (**D**) are labeled. Images in (**B**–**D**) were created in DS ViewerPro 6.0 (Dassault Systèmes, San Diego, CA, USA). (**E**–**H**) The binding of indicated mutants of WT arrestin-1 to P-Rh* (**E**) and Rh* (**F**), as well as the binding of truncated (Tr) arrestin-1-(1–378) with the same mutations to P-Rh* (**G**) and Rh* (**H**), were determined using radiolabeled arrestins produced in cell-free translation in the direct binding assay with purified phosphorylated or unphosphorylated light-activated bovine rhodopsin, as described in Methods. Circles represent measurements in independent experiments (n = 3), each performed in duplicate. The binding was analyzed separately in each of the four groups. Statistical significance of the differences between parental WT (**E**,**F**) or Tr (**G**,**H**) (darker shaded bars in panels (**E**–**H**)) and mutants was determined by ANOVA followed by Dunnet post hoc test with correction for multiple comparisons and is indicated, as follows: * *p* < 0.05; ** *p* < 0.01; *** *p* < 0.001. Bars showing statistically significant increases and decreases in binding are shaded light green and pink, respectively.

**Figure 4 ijms-24-08903-f004:**
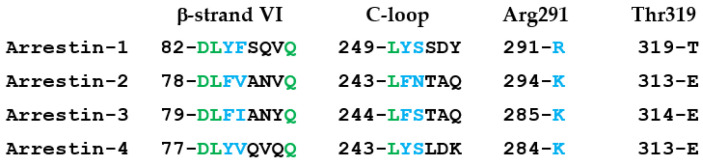
**Sequence comparison of the four arrestin subtypes.** The alignment of the sequences of β-strand VI, C-loops, and other arrestin elements of bovine arrestin-1, -2, -3, and -4. The numbers of the first residue are indicated. Conserved residues are shown in green, residues with conservative substitutions are in blue, and non-conserved residues in black.

**Table 1 ijms-24-08903-t001:** **The effect of mutations on rhodopsin binding.** The arrows indicate the direction of change and the number of stars indicates statistical significance as follows: *, *p* < 0.05; **, *p* < 0.01; ***, *p* < 0.001. Arrows up (↑) and down (↓) indicate statistically significant increases and decreases in binding, respectively. Double arrow down (↓↓) indicates a dramatic (more than 50%) reduction.

Mutation	WT/P-Rh*	WT/Rh*	Tr/P-Rh*	Tr/Rh*
D82A		↑ ***		↓ *
D82R	↑ ***	↑ ***	↓ ***	↓ *
L83A	↑ **			↓↓ ***
Y84A		↑ *		
Y84F				
F85A			↑ ***	
S86A	↑ *			
Q87A		↑ **		↑ ***
V88A			↓ ***	↓ **
Q89A	↑ *		↓ ***	↓ ***
L249A	↓ *	↓ **	↑ ***	↓ ***
Y250A	↓ *	↓ ***	↑ **	↓ ***
S251A		↓ ***		↓ **
D253A			↓ ***	↓ ***
D253R			↓ ***	↑ ***
Y254A				↓ *
Y254F				
R291A	↑ ***	↑ ***	↑ **	↑ ***
R291E	↑ *			↑ ***
T319A				
T319E	↓ ***	↓ ***	↑ **	↑ ***

## Data Availability

The data are presented in the manuscript. Raw binding data obtained in each experiment are available upon request.

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
