# Peer review of "Functional Role of Arrestin-1 Residues Interacting with Unphosphorylated Rhodopsin Elements"

_ijms, 2023, doi:10.3390/ijms24108903_

Round 1

Reviewer 1 Report

Vishnivetskiy et al. investigate the role of single residues of arrestin 1 to mediate selectivity for light-activated phosphorylated rhodopsin. They focus on such residues that are in structural proximity to rhodopsin but are not part of the two common structural elements, the “activation sensor” and the “phosphorylation sensor”, of arrestins. To perform their analysis, they measure interaction of single point mutants of arrestin 1 obtained from in vitro translation with purified rhodopsin in different activation states.

This is a straight-forward study which is well written and clearly presented.

Concerns:

The basis of this manuscript are protein-protein interaction measurements at single concentration. Given the statistically significant but often subtle differences in binding, it would be important to know more about the stoichiometry of the binding partners. Are the values reported reflecting Bmax values or are we looking at subsaturation binding? A binding curve for at least WT and Tr arrestin with Rh* and P-Rh* should be included. In the same context, how do authors check protein integrity of arrestin mutants after in vitro translation? Do mutations affect arrestin stability and if so, could that attribute to different binding values (e.g for R291)? Some kind of quality control for the arrestin mutants (e.g. SEC) should be provided.

While the analysis is thorough, interpretation of the data at some points is a bit lengthy and overstating. In general, it is not surprising that also residues other than those of the activation sensor and phosphorylation sensor also can contribute to rhodopsin binding and also to state selectivity, and thus modulate arrestin function. It is not uncommon that protein-protein interactions that involve multiple interaction sites have not evolved for maximal affinity at a single site (especially in signaling proteins that rely on transient interactions). E.g. for Arg291 one could also argue, that it simply reduces the affinity of arrestin for active Rh, and thus the contribution of the phosphorylation sensor for binding becomes more important. From this, I cannot see how it would “call for a modification of the widely accepted model of arrestin-receptor interaction” as stated in the abstract.

Minor comments:

-       Description of arrestin translation and transcription and rhodopsin purification are insufficient. Please provide more details of the methods.

- Fig. 1: Red and magenta colors are hard to discriminate.

 -       Titles of panels 2c-f & 3e-g are confusing. It is not clear whether WT/ Tr refers to rhodopsin or arrestin.

 -       Fig.4 is lacking colors.

 -       Reference 6 contains spelling errors.

Author Response

The basis of this manuscript are protein-protein interaction measurements at single concentration. Given the statistically significant but often subtle differences in binding, it would be important to know more about the stoichiometry of the binding partners. Are the values reported reflecting Bmax values or are we looking at subsaturation binding? A binding curve for at least WT and Tr arrestin with Rh* and P-Rh* should be included.

We thank the reviewer for bringing these important points. Direct binding assay, as performed, is a one-point binding at the concentration that is unlikely to represent the saturation point. Thus, the values do not reflect Bmax. Unfortunately, the affinity of arrestin-1 for P-Rh* can only be determined by indirect methods, as the rate of diffusion of arrestin-1 would require at least 30 min incubation to reach equilibrium, whereas light-activated rhodopsin decays to opsin with single-digit minute kinetics. For this reason, reported affinity values vary widely, and conventional saturation curve cannot be obtained. The reviewer is right that lower single-point binding reflects reduced affinity for P-Rh*, although this is an indirect assessment.

In the same context, how do authors check protein integrity of arrestin mutants after in vitro translation? Do mutations affect arrestin stability and if so, could that attribute to different binding values (e.g for R291)? Some kind of quality control for the arrestin mutants (e.g. SEC) should be provided.

The reviewer brings a very important point. We added the following explanation to methods:

Upon completion of translation, the mix was cooled on ice and centrifuged for 1 h at 100,000 rpm (>100,000xg) in TLA 100.1 rotor in Optima TLX tabletop ultracentrifuge (Beckman) to pellet ribosomes along with misfolded/aggregated proteins. Protein-incorporated radioactivity was determined before and after high-speed centrifugation. The fraction of the radiolabeled protein that remains in the supernatant after centrifugation was determined and used as a criterion of proper protein folding and stability (it was 80-85% for WT arrestin-1 and all mutants used in this study). 

We do not use mutants that yield significantly lower than WT arrestin-1 fraction of soluble protein, as low yield indicates problems with folding and/or stability.       

While the analysis is thorough, interpretation of the data at some points is a bit lengthy and overstating. In general, it is not surprising that also residues other than those of the activation sensor and phosphorylation sensor also can contribute to rhodopsin binding and also to state selectivity, and thus modulate arrestin function. It is not uncommon that protein-protein interactions that involve multiple interaction sites have not evolved for maximal affinity at a single site (especially in signaling proteins that rely on transient interactions). E.g. for Arg291 one could also argue, that it simply reduces the affinity of arrestin for active Rh, and thus the contribution of the phosphorylation sensor for binding becomes more important. From this, I cannot see how it would “call for a modification of the widely accepted model of arrestin-receptor interaction” as stated in the abstract.

We appreciate reviewers’ point of view. Indeed, individual sites in the multi-site protein-protein interactions do not necessarily exhibit maximum possible affinity. The reviewer is right that lower binding indicates lower global affinity. As the widely accepted model qualitatively explains high global affinity for the preferred by arrestin-1 form of rhodopsin, P-Rh*, the ability of certain mutations to increase the binding to P-Rh* suggests that the native residues in certain positions do not support the binding, but rather impede it. The only other biologically important property of arrestin-1 is its selectivity for P-Rh* over all other forms of rhodopsin, including Rh*. Thus, we concluded that these residues increase arrestin-1 selectivity. This conclusion is supported by the finding that all mutations increasing the binding to P-Rh* increase the binding to the non-preferred form, Rh*, to a greater degree, thereby making arrestin-1 less selective. As this concept is not intuitive, it necessitated lengthy explanations.

Minor comments:

-       Description of arrestin translation and transcription and rhodopsin purification are insufficient. Please provide more details of the methods.

Thank you! We expanded relevant part of methods. However, we did not describe transcription and translation in excruciating detail, as this was done previously in the references cited.

- Fig. 1: Red and magenta colors are hard to discriminate.

Thank you! In Fig. 1 we changed magenta to blue for better discrimination.

 -       Titles of panels 2c-f & 3e-g are confusing. It is not clear whether WT/ Tr refers to rhodopsin or arrestin.

Thank you! As the space in the figure itself is limited, we provided additional explanation in the legend.

 -       Fig.4 is lacking colors.

Thank you for noticing this error. The colors mentioned in the legend were restored.

 -       Reference 6 contains spelling errors.

Thank you for pointing this out. This reference was corrected.

Reviewer 2 Report

This paper concerns a systematic analysis of the effects of site-directed mutagenesis of amino acid residues in arrestin-1 on its binding to light-activated and phosphorylated light-activated whole (WT-P-Rh* and WT-Rh*) and truncated rhodopsin (Tr-P-Rh* and Tr-Rh*). The authors rightly point out that it is difficult to compare the results of a number of previous studies because the effect of mutations was assessed using different parameters, not always directly by the binding of the two partners. In addition to the standard replacement with an alanine residue, replacements were also made with amino acid residues opposite in charge as well as with hydrophobic amino acid residues. The results obtained are summarized in Table 1. Contrary to expectations, in some cases the affinity between rhodopsin and arrestin-1 increased as a result of the mutations. On the whole, the data obtained as a result of the study confirm the authors' main conclusion that the existing model of interaction between arrestin-1 and rhodopsin should be revised. The layout of the manuscript is slightly spoiled by the transfer of the figure legends to the following pages.

Author Response

This paper concerns a systematic analysis of the effects of site-directed mutagenesis of amino acid residues in arrestin-1 on its binding to light-activated and phosphorylated light-activated whole (WT-P-Rh* and WT-Rh*) and truncated rhodopsin (Tr-P-Rh* and Tr-Rh*). The authors rightly point out that it is difficult to compare the results of a number of previous studies because the effect of mutations was assessed using different parameters, not always directly by the binding of the two partners. In addition to the standard replacement with an alanine residue, replacements were also made with amino acid residues opposite in charge as well as with hydrophobic amino acid residues. The results obtained are summarized in Table 1. Contrary to expectations, in some cases the affinity between rhodopsin and arrestin-1 increased as a result of the mutations. On the whole, the data obtained as a result of the study confirm the authors' main conclusion that the existing model of interaction between arrestin-1 and rhodopsin should be revised.

We thank the reviewer for these encouraging comments.

The layout of the manuscript is slightly spoiled by the transfer of the figure legends to the following pages.

Thanks! We hope that in the final layout in the journal the legends will fit the page where the figure is presented.

Reviewer 3 Report

These researchers have contributed significantly in the area of interaction between arrestin-1 and rhodopsin. Their research have provided critical information in understanding the role of arrestin-1 structural elements in the detecting the active conformation of rhodopsin and the phosphorylation sensor responsive to the rhodopsin phosphorylation. In the present manuscript the authors have provided evidence showing the mutations by employing site-directed mutagenesis that  the functional residues of the arrestin-1 binding domain to both phosphorylated and light-activated unphosphorylated rhodopsin (Rh*). Many mutations that enhanced the binding of arrestin-1 to nonphosphorylated Rh* have been identified. The results indicate the native residues present in the wild type arrestin-1 functions like a suppressor element that prevents binding to the nonphosphorylated Rh* thus increasing selective binding of arrestin-1 to the phosphorylated Rh*.

Overall this is a well-written manuscript that provides new information on the arrestin-1 binding to the rhodopsin.   

Author Response

Thank you for encouraging comments!

Round 2

Reviewer 1 Report

Authors have adressed all my concerns sufficiently.

I would still suggest to relabel panels of Figs. 2C-F and 3E-H and table 1 to better discriminate the interacting molecules (e.g. "WT/P-Rh*" instead of "WT-P-Rh*" . The current nomenclature rather suggests it to be a single molecule.

Author Response

Corrections in Table 1, as well as in Figs 2 and 3 were made and reflected in respective legends. The legend to Fig 1 was also corrected.